# Degradation of Bisphenol A by *Bacillus subtilis* P74 Isolated from Traditional Fermented Soybean Foods

**DOI:** 10.3390/microorganisms11092132

**Published:** 2023-08-22

**Authors:** Young Kyoung Park, Young-Wook Chin

**Affiliations:** Research Group of Traditional Food, Korea Food Research Institute, Iseo-myeon, Wanju-gun 55365, Jeollabuk-do, Republic of Korea

**Keywords:** bisphenol A, *Bacillus subtilis*, BPA degradation, BPA-contaminated food, fermented soybean

## Abstract

Bisphenol A (BPA), one of the most widely used plasticizers, is an endocrine-disrupting chemical that is released from plastic products. The aim of this study was to screen and characterize bacteria with excellent BPA-degrading abilities for application in foods. BPA degradation ability was confirmed in 127 of 129 bacterial strains that were isolated from fermented soybean foods. Among the strains, *B. subtilis* P74, which showed the highest BPA degradation performance, degraded 97.2% of 10 mg/L of BPA within 9 h. This strain not only showed a fairly stable degradation performance (min > 88.2%) over a wide range of temperatures (30–45 °C) and pH (5.0–9.0) but also exhibited a degradation of 63% against high concentrations of BPA (80 mg/L). The metabolites generated during the degradation were analyzed using high-performance liquid chromatography–mass spectrometry, and predicted degradation pathways are tentatively proposed. Finally, the application of this strain to soybean fermentation was conducted to confirm its applicability in food.

## 1. Introduction

Bisphenol A (BPA) is a synthetic compound that is frequently used in the manufacturing of polycarbonate- and epoxy resin-based plastic products, including water bottles, cups, cans, tableware, food packaging, and food containers [1]. Although BPA gained popularity after its development, it has been the subject of continuous controversy since the 2000s because of its adverse effects on humans and animals. The number of studies dealing with BPA and its potential hazards to organisms has dramatically increased over the past decade.

BPA, with its two benzene rings and two (4,4′)-OH substituents, has a high binding affinity for various cell surface receptors such as estrogen-, androgen-, membrane-, and epidermal growth factor-related receptors [2]. This binding disrupts the endocrine system by altering the genomic and non-genomic signaling pathways in different cell types [3]. BPA has been reported to cause a wide variety of adverse effects, such as cancer [4], obesity, diabetes [5], mutagenesis [6], reproductive toxicity [7], immune-toxicity [8], embryo-toxicity [9], and transgenerational effects [10] in animals and humans.

BPA derived from plastic products is ubiquitous and easily detected in the surrounding environment, including in surface waters, soils, air, and wildlife. BPA has also been found in human serum, blood, plasma, and urine [11,12,13]; it has also been identified in the placenta and in amniotic fluid, indicating the possibility of BPA exposure during fetal development [14]. Moreover, some reports have highlighted the exposure risk of breastfed infants to BPA because it can be excreted in breast milk [15].

There are physical, chemical, and biological methods for removing BPA; bacteria are the most widely used bioremediation agents according to the International Patent Classification [16]. To date, numerous BPA-degrading bacteria, including *Pseudomonas* spp., *Sphingomonas* spp., *Sphingobium* spp., *Cupriavidus* spp., *Achromobacter* spp., and *Acinetobacter* spp., have been isolated from sediments, soil, water, industrial waste, and sludge for the remediation of BPA-contaminated environments [17]. However, as most of the BPA absorbed into the human body originates from food, there is a need to discovering microorganisms that can be applied for food [18]. Accordingly, generally recognized as safe (GRAS) microorganisms that are applicable to food have been recently reported; however, the number of microorganisms is extremely limited. Most of the microorganisms reported were *Bacillus* and lactic acid bacteria, and the BPA degradation ability of *Bacillus* was better than that of lactic acid bacteria (Table 1). The potential of *Bacillus* as a BPA degrader was also revealed in the screening processes of previous studies. Following the screening of twenty BPA-resistant strains from a microorganism collection that was isolated from human feces, Moreno et al. found that 83% of the strains belonged to the *Bacillus* genus [19]. Mohan et al. also reported that among the eight strains selected from livestock house sludge that were able to grow using BPA as the sole carbon source, *B. subtilis* HV-3 demonstrated the best growth rate [20]. In addition, Yamanaka et al. obtained three halo-forming bacteria by spreading a suspension of kimchi, a Korean fermented food, on a BPA-containing agar plate, all of which were identified as *B. pumilus* [21].

Traditional fermented soybean foods, including doenjang and ganjang, have long been consumed as condiments in Korea. As traditional fermented soybean foods are produced through spontaneous fermentation without a starter culture, their microbial communities are very diverse. Among the microbial communities, some *Bacillus* species, including *Bacillus subtilis*, are the most dominant microorganisms and are noted as GRAS [24]. Some *Bacillus* spp. have exhibited health promoting effects such as anti-obesity [25], immune-enhancement [26], and an anti-diabetic effect [27]. In addition, several *Bacillus* species such as *B. subtilis*, *B. coagulans*, and *B. licheniformis* have already been proven safe enough to be used as probiotics [28].

Based on these clues, the aim of this study was to screen and characterize bacteria with excellent BPA-degrading abilities for potential application in foods. A bacterial strain with an excellent BPA removal ability was screened from traditional fermented soybean foods. Afterwards, the effects of the fermentation conditions on BPA degradation were investigated to evaluate the performance of the strain. The metabolites of BPA degradation were detected using high-performance liquid chromatography–mass spectrometry (LC-MS/MS). Finally, the strain was applied for soybean fermentation to assess their BPA-degrading ability in food.

## 2. Materials and Methods

### 2.1. Isolation and Identification of Bacteria

In our previous study, 129 bacterial strains were isolated from traditional Korean fermented soybean foods [29]. The isolated strains were streaked on a tryptic soy agar (TSA; Difco, MD, USA) plate to obtain single colonies, which were incubated at 37 °C for 24 h. Single colonies were cultured in tryptic soy broth (TSB; Difco, MD, USA) at 37 °C for 24 h and stored at −80 °C with glycerol (50%, *v*/*v*). Isolated strains were identified using 16S rRNA gene sequencing using the universal bacterial primer pairs 27F (5′-AGAGTTTGATCCTGGCTCAG-3′) and 1492R (5′-GGTTACCTTGTTACGACTT-3′). Afterwards, the gene sequences were analyzed using the basic local alignment search tool (BLAST) of the National Center for Biotechnology Information (NCBI; http://www.ncbi.nlm.nih.gov/BLAST/) (accessed on 24 April 2021).

### 2.2. Culture Conditions

The screening of bacteria the with highest BPA degradation ability was conducted by measuring the BPA degradation of 129 strains, respectively. The isolates were pre-cultured in a 96-well deep plate dome (Bioneer Corp., Daejeon, Republic of Korea), each containing 1 mL of sterilized TSB medium at 37 °C for 24 h. BPA (Sigma-Aldrich, St. Louis, MO, USA) was dissolved in sterilized TSB to a final concentration of 10 mg/L; a total of 1 mL was dispensed into each well of a new 96-well deep plate dome and inoculated with 10 µL of the pre-cultured cells. After incubation at 37 °C for 48 h, the BPA in the medium was quantified using high-performance liquid chromatography (HPLC).

After the screening process, all experiments were carried out using the *B. subtilis* P74 strain. The investigation of the effects of culture conditions (BPA concentration, temperature, and pH) was also performed in a 96-well deep plate dome. The concentration of BPA in the sterilized TSB was adjusted to 10, 20, 40, 60, 80, and 100 mg/L and incubated at 30, 35, 37, and 40 °C for 48 h, respectively. For the test on pH effects, the TSB medium was adjusted to pH 4.0, 5.0, 6.0, 7.0, 8.0, and 9.0 using 2 M of hydrochloric acid (Sigma-Aldrich) and 2 M of sodium hydroxide (Sigma-Aldrich). After sterilization, the BPA concentration of the media was adjusted to 10 mg/L. After incubation at 40 °C for 48 h, the concentration of BPA in the medium was measured using HPLC.

The degradation performance of *B. subtilis* P74 was tested in a flask culture containing 100 mL of TSB medium with 10 mg/L of BPA, agitation at 220 rpm and at 35 °C and 40 °C for 48 h. Cell growth was measured at an optical density of 600 nm (OD_600_) using a spectrophotometer (Eppendorf Biospectrometer, Hamburg, Germany). The effect of initial cell concentration was also investigated in flask cultures containing 100 mg/L of BPA at 40 °C under the same conditions. The cells were inoculated to adjust the initial OD_600_ to 0.3, 1.0, 3.0, and 10.0, respectively.

### 2.3. Measurement of BPA Degradation

BPA was dissolved in dimethyl sulfoxide (Sigma-Aldrich) to prepare a stock solution of 10,000 mg/L. To construct a standard calibration curve, the BPA stock solution was serially diluted to 0, 2, 4, 6, 8, and 10 mg/L. The BPA-degrading ability of the strains was determined as described in previous studies [23,30]. Acetonitrile (1 mL) was mixed with 1 mL of the culture medium, depending on each culture condition, and centrifuged at 10,000 rpm at 4 °C for 1 min. The supernatants were filtered through a 0.2-μm PTFE syringe filter (Whatman PLC, Kent, UK) and analyzed using an HPLC system (Agilent 1260 Infinity II LC System, Agilent Technologies, Inc., Santa Clara, CA, USA) equipped with a fluorescence detector (FLD; Agilent Technologies, Inc.) and a Zorbax SB-C18 column (Agilent Technologies, Inc.). The mobile phase was composed of 0.1% formic acid (solvent A) and acetonitrile (solvent B), and the flow rate was set at 1 mL/min (solvent A:B = 50:50). The column temperature was set to 40 °C. The excitation and emission wavelengths were set to 226 and 310 nm, respectively. The sample injection volume was 10 μL. The BPA degradation was calculated as follows:BPA degradation (%) = (*W*_0_ − *W*_1_)/*W*_0_ × 100, (1)
Relative BPA degradation = *W*_1_/*W*_0_,(2)
where *W*_1_ and *W*_0_ refer to the BPA concentrations in the experimental group (with inoculation) and control group (without inoculation), respectively.

### 2.4. Detection of Metabolites Generated from BPA Degradation

An HPLC-MS/MS system (HPLC; Ultimate 3000, Thermo Scientific, Waltham, MA, USA) equipped with a UV detector (Dionex AD20, Dionex, CA, USA) at 365 nm, a CORTECS C18 column (2.1 mm × 150 mm, 1.6 μm) at 45 °C, and mass spectrometer (TripleTOF 5600+ System, AB Sciex, CA, USA) was used to detect the metabolites generated from BPA degradation in the sample. A mixture of 10 mM of ammonium acetate in water and acetonitrile at a ratio of 25:75 with a flow rate of 0.35 mL/min was taken as the mobile phase. The electrospray ionization (ESI) was used at an ion spray voltage of 4.5 kV and temperature of 500 °C.

### 2.5. Soybean Fermentation

Washed soybeans (Glycine max Merrill) were soaked in distilled water for 12 h. After removing the distilled water, the soaked soybeans were drained for 1 h. Subsequently, 200 g of soybeans were placed in flasks and steamed at 121 °C for 30 min in an autoclave. After cooling the steamed soybeans, BPA was added up to a concentration of 10 µg/g. The pre-cultured *B. subtilis* 168 (ATCC33234, type strain) was used as a control and *B. subtilis* P74 was inoculated to 1% (*v*/*w*) and fermented at 40 °C for 48 h.

After the fermentation, 1% (*w*/*v*) formic acid (10 mL) and acetonitrile (10 mL) were added to 5 g of fermented soybean sample, and the mixture was homogenized for 1 min using a homogenizer (IKA^®^-Werke GmbH & Co. KG, Staufen, Germany). To purify the samples, a QuEChERS extraction kit (Agilent Technologies, Inc.) was used, and the mixture was vigorously shaken for 1 min, followed by centrifugation at 3500 rpm at 4 °C for 10 min. After that, 1 mL of the supernatant was transferred into a QuEChERS dispersive kit (Agilent Technologies, Inc.), vortexed for 1 min, and centrifuged at 10,000 rpm at 4 °C for 5 min. The supernatant was filtered through a 0.2 μm PTFE syringe filter and analyzed using HPLC. The mobile phase was composed of 0.1% formic acid in water (solvent A) and acetonitrile (solvent B); the ratio of solvent B was 10% for 0–2 min, 50% for 6–9 min, 90% for 11–13 min, and 10% for 14–15 min. The HPLC analysis condition and calculation of BPA degradation were the same as above.

### 2.6. Statistical Analysis

All experimental results are presented as the mean and standard deviation as derived from three measurements. An analysis of variance (ANOVA) was performed, and the significance was determined at *p* < 0.05 using the Tukey honestly significant difference (HSD) test. All statistical analyses were performed using Minitab statistical software, version 17 (Minitab, Inc., State College, PA, USA).

## 3. Results

### 3.1. Screening of Bacteria with High BPA Removal Ability Isolated from Fermented Soybean Products

The 129 bacterial strains isolated from Korean traditional fermented soybean products were identified, and the BPA removal of each isolate was evaluated in order to select a strain with the highest BPA-degrading ability (Figure 1). Surprisingly, most of the isolates identified as *Bacillus* spp. showed the ability to remove BPA. Among the isolates, *B. subtilis*, *Bacillus* spp., *B. licheniformis*, and *B. amyloliquefaciens* were the most dominant, and their maximum BPA degradations were 14.2–92.1% (51 strains), 10.4–58.0% (23 strains), 6.2–54.5% (14 strains), and 7.6–54.0% (13 strains), respectively (Appendix A). Among these strains, *B. subtilis* P74 showed the highest BPA degradation of 92.1%.

Through the outcome of this study, it was newly revealed that 12 species of *Bacillus* (*B. aerius*, *B. atrophaeus*, *B. coagulans*, *B. glycinifermentans*, *B. licheniformis*, *B. paralicheniformis*, *B. safensis*, *B. siamensis*, *B. sonorensis*, *B. sporothermodurans*, *B. vallismortis*, *B. velezensis*) possess the potential to degrade BPA. Furthermore, the results showed that even within the same species, there was a large difference in BPA removal ability depending on the strain.

### 3.2. Effects of Culture Conditions on BPA Degradation by B. subtilis P74

To characterize the BPA degradation by *B. subtilis* P74, which exhibited the highest BPA degradation ability, the effects of BPA concentration, incubation temperature, pH, and inoculation cell concentration were investigated in flask cultures.

#### 3.2.1. Effects of BPA Concentration and Temperature on BPA Degradation from *B. subtilis* P74

To investigate the effects of BPA concentration and incubation temperature on the BPA degradation by *B. subtilis* P74, a degradation in the range of 10–100 mg/L of BPA was measured at 30–40 °C. As a result, the degradation of BPA was inhibited in a concentration-dependent manner at all temperatures tested (Figure 2a). In particular, at 30 °C, the BPA degradation was reduced to less than half, even at 20 mg/L of BPA, and no degradation was observed at 60 mg/L or more. On the other hand, at 40 °C, a relatively high degradation (≥60%) was shown, even at a BPA concentration of 80 mg/L. When the BPA concentration was 100 mg/L, the BPA degradation was seriously inhibited by less than 10% at all temperatures. In other words, the impact of temperature on BPA degradation was relatively greater at medium concentrations (20–80 mg/L) of BPA than at low (10 mg/L) and high (100 mg/L) concentrations. As an additional test on temperature, we measured the degradation of 10 mg/L of BPA at 25 °C and 45 °C (Figure 2b). As a result, while there was no significant change in the degradation at 45 °C, it was reduced to less than 60% at 25 °C.

#### 3.2.2. Effects of pH on BPA Degradation by *B. subtilis* P74

The effect of initial pH on BPA degradation by *B. subtilis* P74 was investigated by measuring the degradation at the pH range between 4.0 and 9.0. To minimize the influence of other factors, the incubation was carried out at 40 °C with a BPA concentration of 10 mg/L. The results indicated that a high BPA degradation of 91.4−98.0% was exhibited in a wide range of pH 5.0–9.0 (Figure 2c). In contrast, the BPA degradation was significantly low (2.4%) at pH 4.0, which is an acidic condition.

#### 3.2.3. BPA Degradation Performance of *B. subtilis* P74

To evaluate the BPA degradation performance of *B. subtilis* P74 in more detail, the changes in BPA degradation during fermentation were investigated in a flask culture containing 10 mg/L of BPA at 35 °C and 40 °C (Figure 3). We observed that, at 40 °C, 10 mg/L of BPA added to the medium was degraded by 97.2% in 9 h with a degradation rate of 1.08 mg/L/h, which was 34% faster than that at 35 °C.

#### 3.2.4. Effects of Initial Cell Concentration on BPA Degradation by *B. subtilis* P74

From the above results, we confirmed that the BPA degradation by *B. subtilis* P74 was significantly inhibited at concentrations above 60 mg/L; especially, at 100 mg/L, the degradation was reduced to less than 10%, regardless of temperature (Figure 2a). Because the concentration of BPA present in food or in the environment is reported to be less than 1 mg/L [31], even the initial OD_600_ = 0.03 used in the above experiments can be sufficiently removed; however, in harsh environments such as industrial wastewater, high concentrations of BPA may be present. Therefore, we investigated whether a high concentration of BPA could be degraded through high-cell density inoculation. The culture was carried out in a flask culture containing 100 mg/L of BPA, and the cells were inoculated to adjust the initial OD_600_ to 0.3, 1.0, 3.0, and 10.0, respectively (Figure 4a and Appendix A). We observed that, even when the initial OD_600_ was only 0.3, 100 mg/L of BPA was almost degraded within 12 h (BPA degradation = 99.7% ± 0.01). The BPA degradation enhanced as the cell inoculation concentration increased, and when the cell inoculation concentration was OD_600_ = 10.0, 100 mg/L of BPA was completely degraded within 1.5 h. In terms of cell growth, it was clearly observed that the lag phase shortened as the cell inoculation concentration increased (Figure 4b and Appendix A).

### 3.3. Mechanism of BPA Degradation by B. subtilis P74

To confirm whether the BPA removal by *B. subtilis* P74 was due to absorption, dead cells were applied for BPA removal. The results indicated that BPA in the medium was not removed by dead cells (Table 2). This is contrary to the result of a previous study and therefore suggests that the mechanism of BPA removal by *B. subtilis* P74 is different from that of *Lactococcus lactis* [32]. Subsequently, intracellular BPA was measured by sonicating the cells used for BPA removal. As shown in Table 2, the concentration of intracellular BPA was not significantly different from that of the extracellular BPA. Therefore, it was confirmed that BPA removal by *B. subtilis* P74 does not occur through adsorption.

Subsequently, the metabolites of BPA degradation were analyzed using LC-MS/MS to investigate the degradation mechanism. As shown in Appendix A, it was clearly confirmed that the peak of BPA (*m*/*z* = 227.1089, RT = 10.08) decreased as the fermentation proceeded. Several candidate metabolites were estimated based on *m*/*z*, chemical formulae, and previous studies (Appendix A): 1,2-bis(4-hydroxyphenyl)-2-propanol, 4-hydroxybenzaldehyde, 4-hydroxyacetophenone, and syringic acid. Based on the result and previous studies, we tentatively propose the BPA degradation pathway to be as shown in Figure 5. BPA is converted to 1,2-bis(4-hydroxyphenyl)-2-propanol through the hydroxylation of its quaternary carbon atom, where it is then cleaved into 4-hydroxybenzaldehyde or 4-hydroxyacetophenone through oxidation. The two types of benzene ring compounds are converted to catechol and hydroquinone via 3,4-dihydroxybenzoic acid and 4-hydroxyphenylacetate, respectively. The following reactions with enzymes or radicals causes aromatic ring cleavage, which results in the formation of small compounds such as organic acids, acetaldehyde, and acetyl-CoA, which are eventually mineralized through the TCA cycle [33].

### 3.4. BPA Degradation by B. subtilis P74 during Soybean Fermentation

Controlling BPA in food is crucial because humans primarily ingest it through food. To assess the BPA-degrading ability of *B. subtilis* P74 in food, this strain was applied to soybean fermentation. The BPA concentration of the steamed soybeans was adjusted to 10 mg/L; the soybeans were inoculated with *B. subtilis* P74 while using the type strain *B. subtilis* 168 as a control, and the soybeans were fermented at 40 °C for 48 h prior to measuring BPA degradation. The BPA degradation after fermentation was 59.3% in the fermented soybean inoculated with *B. subtilis* P74, which was about 46 times higher than that of the control (Figure 6). However, this BPA degradation was about 30% lower than the degradation measured in the medium, probably because the soybean fermentation was conducted in the solid-state. Nevertheless, *B. subtilis* P74 showed an excellent BPA degradation of nearly 60%, demonstrating its applicability in fermented foods.

## 4. Discussion

The BPA removal ability of 15 species of *Bacillus* (129 strains) isolated from fermented soybean foods was evaluated in this study. The results showed that even within the same species, there was a large difference in BPA removal ability depending on the strain. This strain dependency was also reported for three *B. pumilis* strains that were isolated from kimchi [21]. Among the 15 *Bacillus* species, 12 species were found to have the ability to remove BPA. To the best of our knowledge, the ability of 12 *Bacillus* spp. to degrade BPA, excluding *B. subtilis* and *B. amyloliquefaciens*, is a novel discovery. Because they are all food-derived and because some are already GRAS strains (*B. coagulans*, *B. licheniformis* and *B. subtilis*), they may be a versatile option for food and human applications; however, additional safety studies, including toxicity assessment to living organisms, are needed.

BPA degradation by *Bacillus* spp., which is inhibited in a concentration-dependent manner, has also been reported in some previous studies. The kimchi-derived strain of *B. pumilus* degraded 25 mg/L of BPA within 2 days, although it took 5 days to degrade 50 mg/L and no cell growth or degradation was observed at 100 mg/L of BPA [21]. *B. megaterium* isolated from industrial water waste degraded most of the 5 mg/L of BPA for 72 h; however, the degradation was halved at a concentration of 20 mg/L [34]. The growth of *B. subtilis* was unaffected at BPA concentrations of ≤30 mg/L, whereas the growth was inhibited at ≥40 mg/L [18]. In addition to *B. subtilis*, the growth of *Lactococcus lactis*, *L. plantarum*, *Enterococcus faecalis*, and *Saccharomyces cerevisiae* was also inhibited by increasing the BPA concentrations. It has been reported that BPA is toxic to microorganisms and inhibits their growth, metabolism, and gene expression [35]. BPA also appears to be a similar case to most toxic compounds that exhibit concentration-dependent inhibition.

*B. subtilis* P74 showed fairly stable degradation performance in the wide pH range of 5.0–9.0 (Figure 2c), whereas it was quite weak under the acidic condition (<pH 4.0). These might be due to the effect of pH on the growth of *B. subtilis* and solubility of BPA. Bergey’s Manual states an active growth of *B. subtilis* over a pH range of 6–8 [36]. However, the growth of *B. subtilis* are limited below pH 4.8 [37]. A previous study has also reported that BPA degradation by *B. subtilis* is more inhibited under the acidic condition than under the alkaline condition. The BPA degradation of *B. subtilis* isolated from livestock sludge was 95% and 80% at pH 9.0 and 11.0, respectively, whereas at pH 3.0, it was significantly inhibited to 30% [20]. It has been described that acidic conditions reduce the solubility of BPA, which can affect the BPA degradation, whereas alkaline conditions reduce the BPA degradation by inhibiting bacterial growth [20,38].

Because BPA degradation performance greatly varies with the microbial species, especially with respect to the strain, it is important to discover excellent strains that can lead to an efficient BPA degradation process. Among the 129 strains, *B. subtilis* P74, which showed the best BPA degradation performance, degraded 97.2% of 10 mg/L of BPA within 9 h. This excellent strain not only showed a fairly stable degradation performance (min > 88.2%) over a wide range of temperatures (30–45 °C) and pH (5.0–9.0) but also exhibited a degradation of 63% against high concentrations of BPA (80 mg/L). Table 1 summarizes previous studies on microorganisms with BPA degradation potential that are applicable to food.

In terms of the mechanism, BPA removal by bacteria may be caused via adsorption on the bacterial surfaces by electrostatic forces or chemical affinity [32], as well as through enzymatic degradation [39]. In a previous study, it was proven that BPA removal by *Lactococcus lactis* was due to adsorption by confirming BPA removal by dead cells [32]. In contrast, BPA removal by *B. subtilis* P74 was found to be degradation rather than absorption. Various degradation intermediates of BPA have been identified in several studies so far, and although some microbial degradation pathways have been proposed, the mechanisms are still unclear. Similar to most previous studies, only a few intermediates that are involved in the degradation pathway were detected in this study. This is probably due to the difference in the reaction rate of each step and the presence of many unstable metabolites. Nevertheless, this study detected the metabolites of a single benzene ring compound that was produced by the degradation of BPA, which is composed of two benzene rings. It has been reported that the endocrine-disrupting effects of BPA are caused by the structure of two benzene rings with (4,4’)-OH substituents [2,3]. It has also been reported that the monomers produced through BPA biodegradation have very low acute toxicity compared with BPA and that biodegradation can significantly reduce the toxicity of BPA contaminants [40]. Therefore, BPA degradation by *B. subtilis* P74 is expected to greatly reduce its toxicity. Based on the LC-MS/MS analysis and previous studies, the BPA degradation pathway was tentatively proposed. Similar degradation pathways have also been reported in previous studies [19,20,41]. 

The enzymes involved in the microbial degradation of BPA vary depending on the genus or species, though three enzymes have mainly been reported to play an key role: (i) monooxygenases (e.g., cytochrome P450 and ammonia monooxygenase), (ii) oxidases (e.g., laccase and manganese oxidase), and (iii) peroxidases (e.g., lignin and manganese peroxidases) [33,35,42,43]. In the future, it is necessary to further study the enzymes and genes that are involved in the excellent BPA degradation performance of *B. subtilis* P74 through whole genome sequencing.

## 5. Conclusions

This study demonstrated that *Bacillus* spp. isolated from Korean traditional fermented soybean products show high BPA degradation ability. Among the isolated strains, *B. subtilis* P74 exhibited the highest BPA degradation, and its degradation ability was stable over a wide temperature and pH range. The reduction of BPA by *B. subtilis* P74 was also confirmed in soybean fermentation. These results reveal the applicability of *B. subtilis* P74 in fermented foods, and it is expected that it can be used in other foods or in bioremediation applications. In addition, because *B. subtilis* can be used as a probiotic, research on its related properties is ongoing.

## Figures and Tables

**Figure 1 microorganisms-11-02132-f001:**
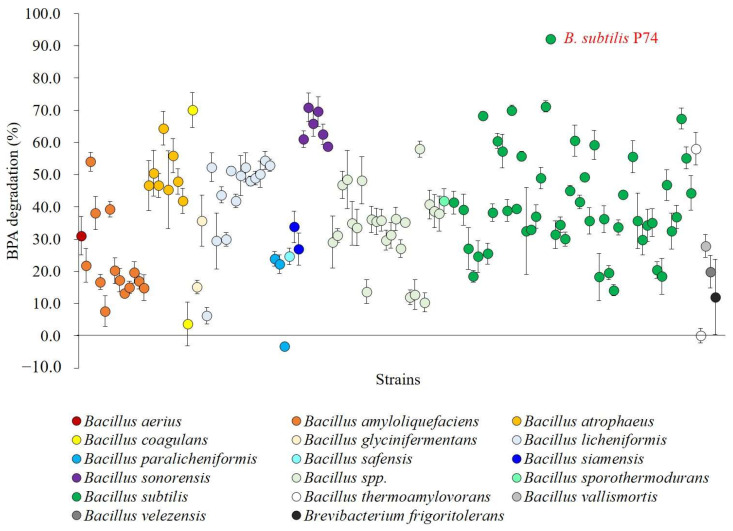
BPA degradation of 129 bacterial strains isolated from fermented soybean products. The data represent the mean ± standard deviation from three independent experiments.

**Figure 2 microorganisms-11-02132-f002:**
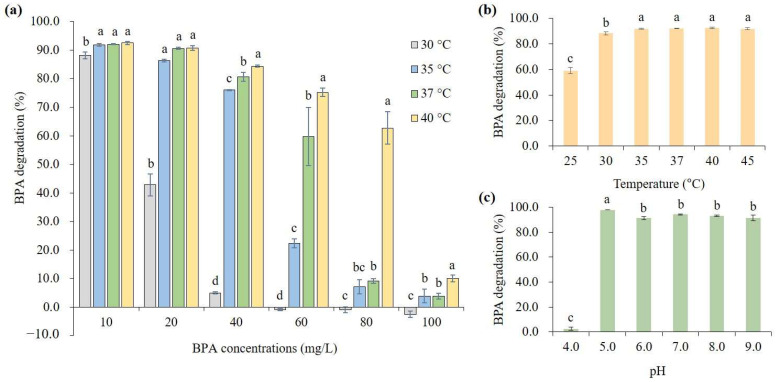
Effects of culture conditions [(**a**) BPA concentration; (**b**) temperature; (**c**) pH] on BPA degradation by *B. subtilis* P74. The data represent the mean ± standard deviation from three independent experiments, and the values with different letters on the bar indicate significant differences (*p* < 0.05).

**Figure 3 microorganisms-11-02132-f003:**
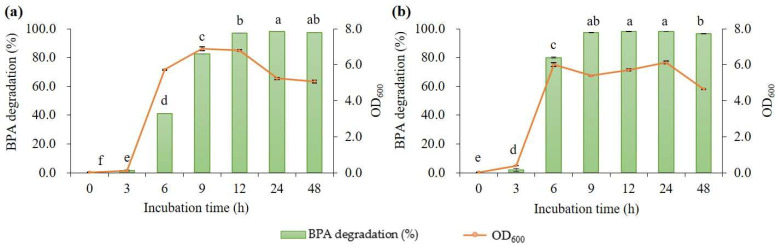
BPA degradation performance in a flask culture containing 10 mg/L of BPA by *B. subtilis* P74 at (**a**) 35 °C and (**b**) 40 °C. The data represent the mean ± standard deviation from three measurements of harvested samples, and the values with different letters on the bar indicate significant differences (*p* < 0.05).

**Figure 4 microorganisms-11-02132-f004:**
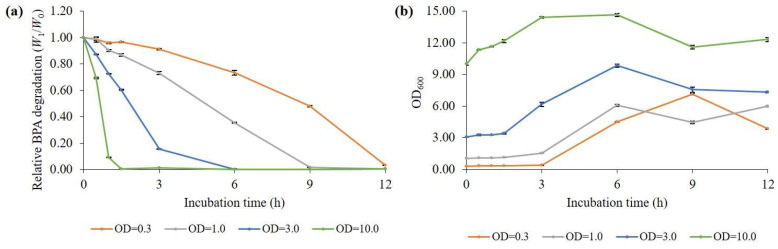
Effects of initial cell concentration on (**a**) BPA removal efficiency and (**b**) cell growth in a medium containing 100 mg/L of BPA at 40 °C. The data represent the mean ± standard deviation from three measurements of harvested samples.

**Figure 5 microorganisms-11-02132-f005:**
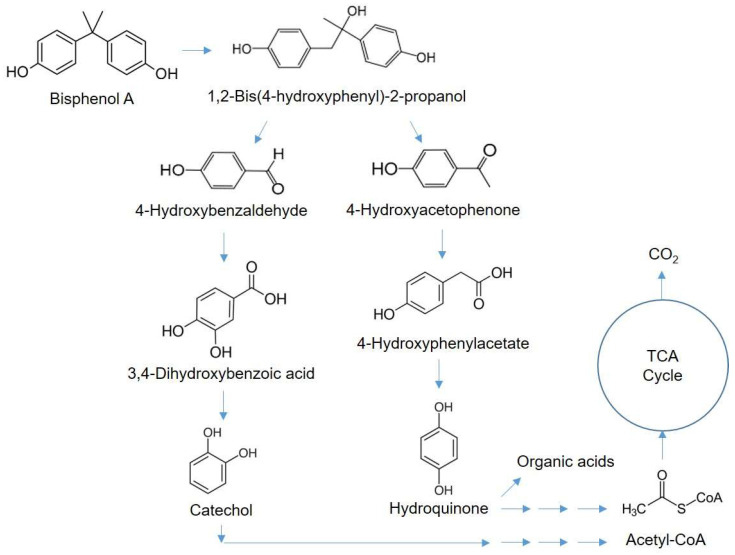
Predicted pathway of BPA degradation by *B. subtilis* P74.

**Figure 6 microorganisms-11-02132-f006:**
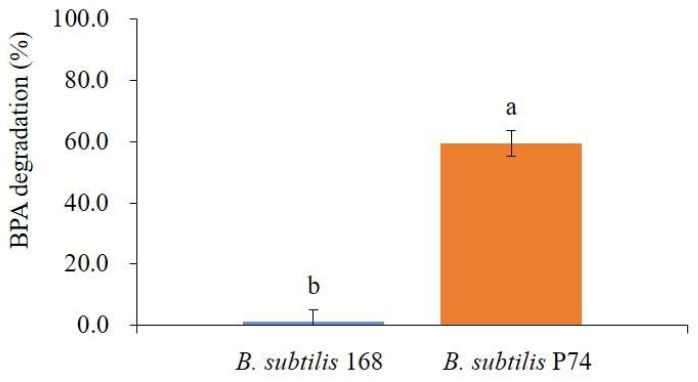
BPA degradation in soybean fermentation inoculated with *B. subtilis* 168 and *B. subtilis* P74. The data represent the mean ± standard deviation from measurements of samples harvested in triplicate, and the values with different letters on the bar indicate significant differences (*p* < 0.05).

**Table 1 microorganisms-11-02132-t001:** Summary of previous studies on GRAS bacteria capable of degrading BPA.

Microorganism	Source	BPA Degradation Performance ^a^	Reference
*B. pumilus* BP-22DK	Kimchi	10 mg/L (100%), 16 h	[21]
*B. subtilis* HV-3	Livestock house sludge	100 mg/L (98%), 120 h (electrochemical system)	[20]
*B. amyloliquefaciens* SBUG1837	Activated sludge	60 mg/L (77%), 48 h	[22]
*B. subtilis*	Soil	50 mg/L (52%), 96 h	[18]
*Lactococcus lactis* RKG1-319	Dairy product	50 mg/L (39%), 96 h	
*Lactobacillus plantarum* 2035	Dairy product	50 mg/L (42%), 96 h	
*Enterococcus faecalis* ATCC 19433	Lab strain	50 mg/L (45%), 96 h	
*L. reuteri* ATCC55730	Lab strain	1 mg/L (70%), 48 h	[23]

^a^ BPA degradation performance was expressed as BPA concentration (degradation %) and degradation time.

**Table 2 microorganisms-11-02132-t002:** Concentration of residual BPA after flask culture containing 10 mg/L of BPA.

	Control (Extracellular)	Dead Cells	Intracellular
Residual BPA (mg/L)	0.27 ± 0.02 ^a^	9.88 ± 0.04 ^b^	0.28 ± 0.02 ^a^

Superscript alphabets indicate significant differences (*p* < 0.05).

## Data Availability

The data presented in this study are available on request from the corresponding author.

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
