# Peer review of "Degradation of Bisphenol A by Bacillus subtilis P74 Isolated from Traditional Fermented Soybean Foods"

_microorganisms, 2023, doi:10.3390/microorganisms11092132_

Round 1

Reviewer 1 Report

Dear authors,

The subject of research is interesting and specific because concerns the study of pollutant-degrading microbial strain potentially suitable for use in the food industry.

However, before recommendation this manuscript for publication I would like to clarify some points given below. In my opinion, it is important  to carefully structure the manuscript, in particular, the Results section should only include the presentation of the results obtained, while now it includes the discussion (reasoning, comparison with other published data) and looks like the "Results and Discussion" followed by separate “Discussion” section.

Materials and Methods

Section 2.2., line 98. What was an initial optical density of cells at 600 nm in the first experiments before studying the effect of initial cell concentration on bisphenol degradation?

Section 2.4. In this section authors describe rather detection than identification of bisphenol metabolites. Could authors provide some information about reference compounds or metabolite data base on the basis of which the detected intermediates formed from bisphenol have been identified.

Results

This section named “Results”, but not “Results and Discussion”, it should include only the presentation of the results obtained, while now it includes the discussion (reasoning, comparison with other published data) and looks like the "Results and Discussion" section. Please structure the manuscript more clearly by moving the discussion to the section “Discussion”.

The comparison of the results obtained with other reported data should be given in “Discussion” section. For example: Section 3.1., lines 185-186.; Section 3.2., lines 190-191; Section 3.2.1. 212-223; Section 3.2.2., lines 232-243; Section 3.3., lines 276-280; 299-303

Section 3.2.4. How were the initial optical densities used (0.3, 1.0, 3.0, and 10.0) corresponded to initial cell concentrations in previous experiments?

Section 3.3., lines 290 and followed. How were the bisphenol metabolites identified?

Section 3.4., lines 309: “The BPA concentration of the steamed soybeans was adjusted to 10 mg/L.” What for? What could be a real concentration of bisphenol in food?

Discussion

In my opinion, in this section the discussion of the data obtained should be completed and aligned with the discussion given in the “Results” section.

Reviewer 2 Report

The article “Degradation of bisphenol A by Bacillus subtilis P74 isolated from traditional fermented Soybean Foods” by Young Kyoung Park and Young-Wook Chin is devoted to the degradation of bisphenol A by B. subtilis strain P74. It is important to note that the authors of the study attempted to propose the degradation pathway of the said compound. In addition, it was confirmed that the removal of bisphenol A from the medium was not due to adsorption of the substance on bacterial surfaces. The article is detailed and written in competent scientific language. 

The paper https://www.sciencedirect.com/science/article/pii/S1878535223003180 describes that "... an increase in ecotoxicity was observed during the initial stages of bisphenol A (BPA) oxidation. There was a significant correlation between the toxicity patterns and its major transformation products such as benzoquinone, hydroquinone, styrene, p-isopropylphenol, etc.".  Is it possible to use such strains of microbial degraders directly in food? This work is clearly fundamental, given the results presented. Perhaps we should not focus so much on the applied significance of the work, because the mechanism of BPA degradation is not clarified. For example, was the toxicity of the B. subtilis P74 strain assessed in relation to living organisms (since the authors talk about food in their publication)?

There are a number of important comments on the article.

The abstract is concise and fully reflects the purpose and results of the paper.

Introduction

Authors should clearly state the aim of the study. For example, the aim is well written in the abstract. It should be stated at the end of the introduction, before the last paragraph by analogy

Materials and methods

The materials and methods section is detailed. But there are also questions and recommendations:

1) Section 2.1. states that a temperature of 37°C was used. It is recommended to clarify that this is the optimal temperature for bacilli.

2) Section 2.2 states that TSB medium with added BPA was used. Would like to clarify why the authors did not use mineral medium with BPA as the sole source of carbon and energy? Did they perform an experiment to culture the strain in TSB medium without additives? It is very likely that the strain is able to grow in this medium without additives.

3) In section 2.5 the authors cite B. subtilis strain 168, it should be stated why this was used as a control.

4) What number of cells corresponds to the OD given in the article when measuring bacterial cell density?

5) The degradation rate is given in %, this is not correct - the rate is measured e.g. in mg/hour. And in fact, according to the formula in the Materials and Methods section, this index represents the degradation efficiency of BPA?

Results

1.         Are the values in Figure 1 and Figure 2 negative? Can the BPA degradation efficiency (%) be negative?

2. line 258 - Why was a culture temperature of 40°C used to determine the effect of initial cell concentration on bisphenol A removal efficiency and cell growth in BPA-containing medium when it is stated that "degradation rate decreased to less than 10% regardless of temperature"?

3.         What was the abundance of B. subtilis P74 when the BPA degradation efficiency was measured under different conditions?

4.         Figure 3 - How can the decrease in OD after 24 and 48 hours of growth be explained? Also, in this figure the labels of the symbols - f, e, a, b, c, d are missing.

5.         On the ordinate axis of Fig. 4, units of measurement are presented that the authors interpret as removal efficiency, which clearly contradicts the formula presented in the Materials and Methods section. The parameter presented is unclear - either remove this figure so as not to confuse the reader, or include the formula in the Materials and Methods section.

6.         Figure 3 and Figure 4: No standard deviation for growth curves measured by OD600.

7.         Line 276 - probably shouldn't mention fungi as the article is about BPA degrading bacteria?

The Discussion and Conclusion sections are well written and fully capture the essence of the study. Table 2 should remove the fungi and focus only on the bacteria.

References

More than half of the references are less than 5 years old. The references are arranged according to the requirements of the journal. However, the doi of the articles should be entered.  Reference 34 - what an inaccuracy in giving the information. Remove the extra space before the references section.

Minor editing of English language required

Round 2

Reviewer 2 Report

The authors have taken into account all the comments I have recommended for publication.